

# A new, three-dimensional geometric morphometric approach to assess egg shape

Marie R.G. Attard[1,2,*], Emma Sherratt[1,3,*], Paul McDonald[1], Iain Young[1,4], Marta Vidal-García[5] and Stephen Wroe[1]

[1] Zoology Department, School of Environmental and Rural Science, University of New England, Armidale, NSW, Australia
[2] Department of Animal and Plant Sciences, University of Sheffield, Sheffield, South Yorkshire, UK
[3] Department of Ecology and Evolutionary Biology, School of Biological Sciences, The University of Adelaide, Adelaide, SA, Australia
[4] School of Life and Environmental Sciences, Faculty of Science, University of Sydney, Sydney, NSW, Australia
[5] Ecology and Evolution, Research School of Biology, Australian National University, Canberra, ACT, Australia
* These authors contributed equally to this work.

Corresponding authors
Marie R.G. Attard,
marie.r.g.attard@gmail.com
Emma Sherratt,
emma.sherratt@gmail.com

## ABSTRACT

This paper proposes a new methodology to quantify patterns of egg shape variation using geometric morphometrics of three-dimensional landmarks captured on digitally reconstructed eggshells and demonstrates its performance in capturing shape variation at multiple biological levels. This methodology offers unique benefits to complement established linear measurement or two-dimensional (2D) contour profiling techniques by (i) providing a more precise representation of eggshell curvature by accounting for variation across the entire surface of the egg; (ii) avoids the occurrence of correlations from combining multiple egg shape features; (iii) avoids error stemming from projecting a highly-curved three-dimensional (3D) object into 2D space; and (iv) enables integration into 3D workflows such as finite elements analysis. To demonstrate, we quantify patterns of egg shape variation and estimate morphological disparity at multiple biological levels, within and between clutches and among species of four passerine species of different lineages, using volumetric dataset obtained from micro computed tomography. The results indicate that species broadly have differently shaped eggs, but with extensive within-species variation so that all four-focal species occupy a range of shapes. Within-species variation is attributed to between-clutch differences in egg shape; within-clutch variation is surprisingly substantial. Recent comparative analyses that aim to explain shape variation among avian taxa have largely ignored potential biases due to within-species variation, or use methods limited to a narrow range of egg shapes. Through our approach, we suggest that there is appreciable variation in egg shape across clutches and that this variation needs to be accounted for in future research. The approach developed in this study to assess variation in shape is freely accessible and can be applied to any spherical-to-conical shaped object, including eggs of non-avian dinosaurs and reptiles through to other extant taxa such as poultry.

# INTRODUCTION

Avian egg shape diversity is a well-known biological phenomenon reflecting taxonomic diversity (*Olsen, Cunningham & Donnelly, 1994*; *Stoddard et al., 2017*), ranging from the nearly spherical eggs of owls (*Hoyt, 1976*), slightly pointed eggs of domestic fowl (*Havlíček et al., 2008*), to the extremely pointed eggs of certain waders, alcids and penguins (*Birkhead et al., 2017*; *Stoddard et al., 2017*). Various hypotheses have been proposed to explain the evolution of specific egg shapes (*Rensch, 1947*; *Andersson, 1978*; *Bain & Solomon, 1991*; *Attard et al., 2017*; *Deeming, 2017*; *Stoddard et al., 2017*), but so far only explain a small proportion of variation in egg shape (*Deeming, 2017*; *Stoddard et al., 2017*), or are applicable to a limited number of species. These findings are also complicated by the use of multiple methods by different studies to evaluate and compare egg shape, with some more comprehensive in their evaluation of shape differences than others (*Havlíček et al., 2008*).

Historically a range of methods have been applied to quantify egg shape; in morphometrics the term 'form' refers to size plus shape, where 'shape' is defined as 'all the geometric information that remains when location, scale and rotational effects are filtered out from an object' (*Kendall, 1977*). Egg shape is often studied by computing ratios from linear distance measurements (summarised by *Preston, 1969*; *Narushin, 2001*; *Havlíček et al., 2008*; *Troscianko, 2014*); the most common linear ratios used are egg elongation and asymmetry. Egg elongation describes the divergence of an ellipse (flattening) from a circle, whereas asymmetry specifies the extent to which one end of an egg is more pointed than the other. Egg elongation is calculated as the ratio of egg length to breadth (maximum diameter) while egg asymmetry is calculated based on the ratio between the distance from the lower vertex of the egg to the point where the polar axis intersects the equatorial axis to egg length, as described by *Deeming & Ruta (2014)*. Using ratios, such as elongation and asymmetry, to describe egg shape are still widely used despite their limitations. For example, two eggs with the same elongation index do not necessarily imply the same shape, because the same ratio may be obtained if the widest point is at the middle of the egg, or towards the base (*Hoyt, 1979*). Consequently, multiple indices have been used recently in combination to obtain a better description of egg shape. However, these variables are likely to be correlated with one another, leading to multiple difficulties in identifying the extent of shape variation between samples and selecting which indices best characterise egg shape for a given species (*Narushin, 2001*).

Variation in egg curvature is another important component of eggshell shape. Eggshell curvature has been typically analysed based on a two-dimensional (2D) profile of the egg along its longitudinal axis (pole to pole), and was superficially estimated in the past based on direct linear measurements and mathematical equations to approximate the profile of an egg (*Hutt, 1938*; *Bonnet & Mongin, 1965*; *Besch, Sluka & Smith, 1968*;

*Carter, 1968*). However, this methodology is often tedious to apply, and it is difficult to determine how closely the calculated curvature mimics the true profile of the egg (*Carter, 1970*). Other methods have been recently developed in an attempt to capture the precise geometry of such contours. That is the case of curve-fitting equations, which have been used to estimate egg shape through 2D contour profiles (*Nedomova, Severa & Buchar, 2009*; *Troscianko, 2014*), with some requiring separate equations for each species (*Preston, 1953*, *1968*; *Smart, 1991*) or were limited to specific shapes (*Baker & Brawn, 2002*). A comprehensive study on avian egg shape variation by *Stoddard et al. (2017)*, which incorporated 1,400 bird species, was forced to emit irregular or extremely pointed or asymmetric eggs owing to limitations in the curve-fitting equations used, thereby constraining the apparent variation in egg shape.

Even though the use of 2D methods to capture egg shape remains widespread today and will likely remain common for many years owing to its simplicity and low cost, our reliance on using 2D outlines to represent the complex three-dimensional (3D) geometry of eggs poses several limitations. Firstly, 2D analyses rely on an assumption of radial symmetry of the eggshell, where the plane of symmetry is considered to remain the same with rotation along the egg's longitudinal axis (*Deeming & Ruta, 2014*). However, assuming complete radial symmetry can be misleading, as it often ignores subtle differences in curvature around the entire eggshell. *Preston (1969)* acknowledged that irregularities in an egg such as creases, ridges or dents (*Preston, 1968*) can reduce the accuracy of certain shape parameter measurements. Instances of extreme deviation from radial symmetry, as documented in the common guillemot *Uria aalgae* (*Birkhead, 2016*), have not allowed these irregular eggs to be included in 2D shape analysis. Secondly, the 2D approach is prone to error associated with aligning a highly-rounded object exactly parallel to the imaging plane (e.g., camera lens for photographs) (*Troscianko, 2014*), and in many cases, no attempt is made to align the eggs (*Stoddard et al., 2017*). The centre of gravity of most eggs is towards the pointed pole, causing this end of the egg to tilt downwards (for blown and fresh eggs) when placed on a horizontal surface in its resting position (*Mao et al., 2007*). The potential error caused by egg misalignment for 2D analysis are expected to be greatest for elongated or pointed eggs, as the outlines formed by these shapes are more susceptible to small deviations in the angle of the egg's longitudinal axis relative to the imaging plane. Careful egg alignment is needed to ensure 2D silhouettes provide an accurate representation of true egg shape (*Attard et al., 2017*) but, undoubtedly, a 2D approach will still entail certain amount of error when quantifying shape variables. Therefore, in order to fully understand the function of various shapes and overcome the limitations listed above, we need a robust and reliable method to quantify the complete naturally occurring range of egg shapes and curvatures in 3D space.

When studying egg shape in a comparative context, it is important that within- and between-clutch variation in egg characteristics are considered (*Garamszegi & Møller, 2010*). Generally, it is assumed that phenotypic variation among taxa is greater than within-species variation (*Felsenstein, 1985*). However, phenotypic plasticity, population differences or measurement errors are all known to contribute to within-species variability

(*Ives, Midford & Garland, 2007*), which can lead to misinformed conclusions if sample sizes are low. A study of five species of bird found that randomly selecting one egg from each clutch to measure length, breadth and curvature around each pole was more representative of a species than sampling all eggs from fewer clutches (*Preston, 1968*). Sampling bias can also occur in studies where all eggs are sampled from different sized clutches. Such an experimental design would be biased in favour of eggs collected from larger clutches (*Preston, 1968*).

Egg shape variation is expected to be low within a species if shape strongly influences the survival of the young (*Clark, Ewert & Nelson, 2001*). Alternatively, if shape is relatively unimportant for egg hatchability and subsequent offspring survival, then large variation in egg shape within a given species might be expected. Egg shape can increase offspring survival by improving egg strength (*Gosler, Higham & Reynolds, 2005*), embryo growth (*Deeming, 2017*), incubation efficiency (*Drent, 1975*; *Rokitka & Rahn, 1987*; *Deeming & Ferguson, 1991*; *Barta & Székely, 1997*; *Mao et al., 2007*; *Šálek & Zárybnická, 2015*), and the detection of brood parasitism (*Underwood & Sealy, 2006*; *Zölei et al., 2012*; *Attard et al., 2017*). In contrast, within-clutch variation in egg shape is predicted to be a reflection of parental investment in each offspring (dependent on the female's breeding condition and experience) (*Coulson, 1963*), or external factors influencing offspring survival (*Briskie & Sealy, 1990*). Since spherical eggs have the smallest surface area of all 3D solids of a given volume, reducing calcium investment in shell production should therefore be advantageous to females that may be calcium limited (*Gosler, Higham & Reynolds, 2005*). Conversely, elongated eggs provide greater volume to facilitate higher energy and nutrient storage capacity for embryo growth (*Pearl & Curtis, 1916*), and thus, provide the hatchling with greater fitness potential (*Rose, Simpson & Manning, 1996*). Thus if natural selection is acting on egg shape, we expect that differences in eggshell shape will reflect adaptations to optimise incubation and/or hatchability (*Hoyt, 1976*).

In this study, micro computed tomography (micro-CT) data was used to generate 3D models, and geometric morphometrics of 3D landmarks was used to quantify egg shape in four Australian bird species (four clutches per sp.): the grey shrike-thrush *Colluricincla harmonica*, red-browed finch *Neochmia temporalis*, spiny-cheeked honeyeater *Acanthagenys rufogularis* and superb fairy-wren *Malurus cyaneus* (Fig. 1A). Our aim was to investigate whether there is greater shape variation within a clutch than between clutches for a given species, and test if egg shape can be used to distinguish between bird species. Using this dataset as a case study, we apply a 3D landmark method to accurately quantify natural variation in egg shape. Our method builds upon recent developments that attempt to precisely measure the 2D contours of the eggshells from digital photographs using Elliptic Fourier analyses and semilandmark-based geometric morphometric methods (*Johnson, Leyhe & Werner, 2001*; *Havlíček et al., 2008*; *Bravo & Marugán-Lobón, 2012*; *Murray et al., 2013*; *Deeming & Ruta, 2014*; *Deeming, 2017*). These latest developments capture the egg outline and turn the contours into 2D shape variables (*Iwata & Ukai, 2002*; *Murray et al., 2013*) however they are often not as accurate as a 3D approach (*Loy et al., 2000*; *Sheets et al., 2006*).

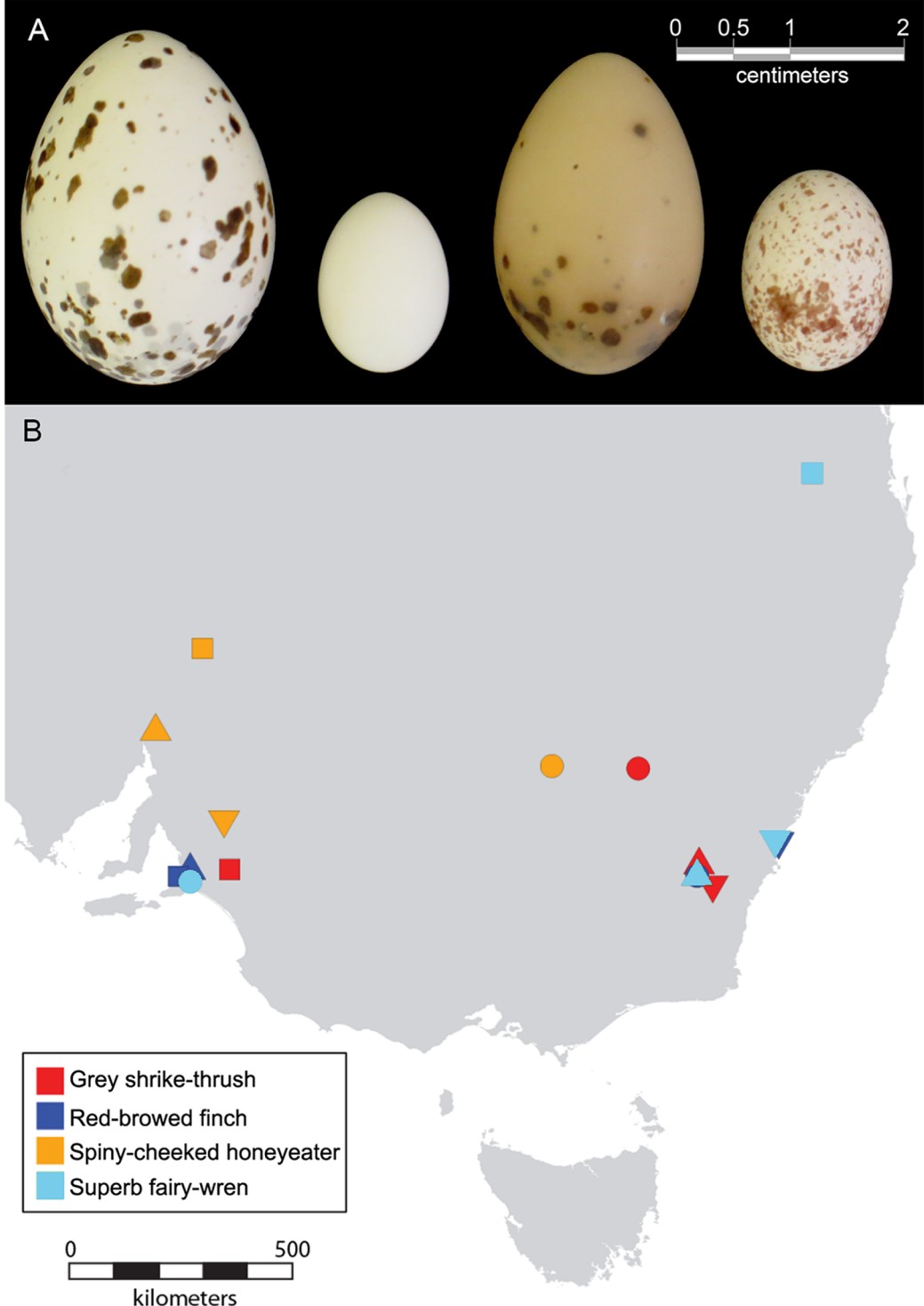

**Figure 1 Photograph of eggs and geographical range of clutches included in this study.** (A) Photographs of eggs from the four species of bird included in this study; from left to right, grey shrike-thrush, red-browed finch, spiny-cheeked honeyeater and superb fairy-wren. (B) Geographical range of bird clutches. See Table 1 for symbol shape used to represent each clutch.

**Table 1 Museum accession and collection information on each clutch included in the study.**

| Common name | Catalogue number | Clutch size | Latitude | Longitude | Date collected | Symbol |
|---|---|---|---|---|---|---|
| Grey shrike-thrush | E14331 | 3 | −34.96 | 149.17 | 14/01/2004 | Triangle |
| Grey shrike-thrush | E14518 | 3 | −35.11 | 139.54 | 06/09/1991 | Square |
| Grey shrike-thrush | E15293 | 2 | −35.42 | 149.45 | 01/10/2006 | Inverted triangle |
| Grey shrike-thrush | E06328 | 3 | −33.01 | 147.92 | 15/09/1999 | Circle |
| Red-browed finch | E06238 | 5 | −35.1 | 138.73 | 27/10/1987 | Triangle |
| Red-browed finch | E10376 | 7 | −35.28 | 138.57 | 08/11/1988 | Square |
| Red-browed finch | E14319 | 5 | −34.57 | 150.77 | 11/12/2003 | Inverted triangle |
| Red-browed finch | E14515 | 5 | −35.22 | 149.13 | 03/01/2005 | Circle |
| Spiny-cheeked honeyeater | E14372 | 2 | −32.19 | 138.02 | 12/09/1988 | Triangle |
| Spiny-cheeked honeyeater | E05185 | 2 | −30.56 | 138.98 | 17/09/1987 | Square |
| Spiny-cheeked honeyeater | E06303 | 2 | −34.1 | 139.43 | 27/08/1992 | Inverted triangle |
| Spiny-cheeked honeyeater | E06324 | 3 | −32.97 | 146.15 | 14/09/1999 | Circle |
| Superb fairy-wren | E10499 | 3 | −35.22 | 149.13 | 15/10/2002 | Triangle |
| Superb fairy-wren | E12643 | 4 | −26.97 | 151.5 | 15/09/2002 | Square |
| Superb fairy-wren | E13865 | 4 | −34.55 | 150.73 | 31/10/1998 | Inverted triangle |
| Superb fairy-wren | E14555 | 3 | −35.34 | 138.69 | 16/11/1992 | Circle |

## METHOD

### Specimens

Egg shape was compared for taxa from four distinct families: the Maluridae (fairy-wrens), Meliphagidae (honeyeaters), Pachycephalidae (whistlers and thickheads) and the Estrildiae (wax bills, grass finches, munias and allies) respectively (Table S1). All four species included, except the grey shrike-thrush, are endemic to Australia. These taxa cover a range of body sizes (10–68 g), diets, clutch size (2–5 eggs per clutch), life histories (pair breeders versus cooperative breeders), and evolutionary origins (Australian Old Endemics versus more recent Eurasian colonists) (Table S1). Within each species, four clutches from different collection locations were preferentially selected to ensure that different clutches were not laid by the same breeding female (Fig. 1B). All clutches within a given species are from the same subspecies, with the exception of the superb fairy-wren, which had one clutch from subspecies *leggei* (E14555), and the others from *cyanochlamys*. All egg clutches were provided by the Australian National Wildlife Collection, Canberra. Collection date, location and clutch size were available for all eggs included in this study (Table 1). Only clutches collected after dichlorodiphenyltrichloroethane (DDT) was banned in 1987 were included to minimise the risk of reported pesticide-induced changes in avian eggshell characteristics (*Fry, 2005*), although, given their dietary preferences, none of these taxa are likely to have been affected by this process in any case. All eggs from each clutch were included in this study, with the exception of one of the seven eggs from clutch E10376, which was too damaged to reliably reconstruct digitally.

## Micro-CT scanning

The intact eggshells ($N = 55$) were scanned using a compact desktop micro-CT scanner (SkyScan 1174; Bruker micro-CT, Kontich, Belgium) at the Australian National Wildlife Collection, Canberra. The following scanning parameters were used: 50 kV source voltage, 80 μA source current, 33.28–33.45 μm pixel size, 360° rotational angle, 0.8° rotational step, 2.3 s exposure time, 40% sharpening, two frame averaging, random movement and flat field correction turned on and a 0.5 mm aluminium filter. The isotropic voxel size used for each scan was specimen specific and was based on the resolution required for the length of the egg to fill 90% of the field of view.

The acquired shadow projections (16-bit TIFF format) were further reconstructed into 2D slices of the structure of each sample using the NRecon software interface (v.1.6.9; Brueker micro-CT, Kontich, Belgium). The reconstruction parameters used were: smoothing (5%), ring artefact correction (20%), beam hardening correction (100%) and setting of contrast limits between 0 and 1.4. A 1.5ml eppendorf tube containing MilliQ water was included in all scans. Micro-CT Hounsfield units (HUs) were calibrated using known water CT density (HU = 0) during 2D slice reconstruction. The cross-section slices were stored in DICOM format and are available on the Figshare repository (https://doi.org/10.6084/m9.figshare.3382477.v1).

## 3D-volume reconstruction

Digital segmentation and solid meshing of micro-CT tomographs was performed with the medical imaging software Mimics (v. 16.0) and 3-matic (v. 8.0), using protocols adapted from *Attard et al. (2014)*. In brief, thresholding and segmentation of egg micro-CT data was conducted in mimics to create a 3D object of the eggshell. The 3D object was imported in 3-matic where the geometry was wrapped, reduced and smoothed, while preserving egg shape and size without data loss. All eggs had been blown cleanly through a hole. To fill in the hole, the surface around the hole of each egg was selected and filled using the freeform tool in 3-matic. The eggshell was separated into two surfaces; one of the interior eggshell surface and one of the exterior eggshell surface. A surface mesh of the exterior eggshell surface consisting of approximately 10,000 triangles was produced for each specimen and exported as a .PLY file for shape analysis, and are available on Figshare (https://figshare.com/s/3af9f0cf5346e9b881f6).

## Quantifying egg shape

We placed 206 landmarks on the 3D surface models to cover the contours of the egg using a digitising routine written in R v.3.2.2 (*R Core Team, 2015*) modified from those in the *geomorph* library (*Adams & Otárola-Castillo, 2013*; *Adams, Collyer & Sherratt, 2016*). Firstly, the egg model was centred and rotated by its principal axes so that the *x*-axis lay along the longest dimension of the egg. The top (pointed pole) and bottom (blunt pole) of the egg were defined by dividing the egg into two halves and calculating the volumes (convex hull volume) of each; the top of the egg is defined as the smaller half. Two landmarks were placed at the poles by taking the minimum and maximum values along the *x*-axis when y is zero. Four landmarks were placed on the *y*- and *z*-axes by the same

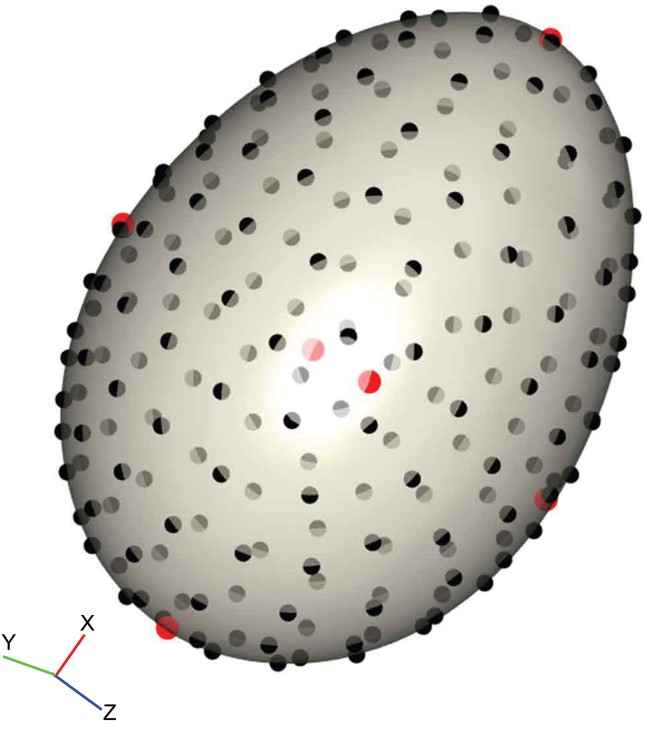

**Figure 2 Example of the digitisation of landmarks on a virtual egg.** The position of six homologous landmarks (pointed pole, blunt pole and four equidistant points around equator) are shown in red, and the template of semilandmarks are shown in black. The surface mesh of the egg was reconstructed from micro-CT data.                         

means, resulting in six landmarks that together demark the height, width and depth, and are positionally homologous on every egg (Fig. 2, red points). Then, 200 equally-spaced semilandmarks were fitted to the first shell surface (Fig. 2, black points) following the algorithm outlined in *Gunz, Mitteroecker & Bookstein (2005)* and *Mitteroecker & Gunz (2009)*, producing a template that could be fitted to every other specimen by thin-plate spline (TPS) warping (*Bookstein, 1989*), using the six polar landmarks to orient the template. This method assures that positional homology is maintained between every semilandmark and works well with curved 3D structures with few homologous landmarks (*Sherratt et al., 2014, 2016*).

The landmark data were aligned using a generalised Procrustes superimposition (*Rohlf & Slice, 1990*); all 206 semilandmarks were permitted to slide in either direction on two planes tangential to the surface in order to minimise bending energy between specimens (*Gunz, Mitteroecker & Bookstein, 2005*). Variation among observations was visualised using a principal components analysis (PCA) and plotting the first two axes in a biplot. The benefit of landmark coordinates is they retain the geometric information and allow biological interpretation of the observed shape variation. We summarised the main shape changes described by the first two PC axes using the surface warp approach (*Drake & Klingenberg, 2010*; *Klingenberg, 2013*; *Sherratt et al., 2014*); an egg model derived from micro-CT reconstruction was warped to the mean shape using the TPS method, and then this reference egg was warped to the shapes representing the

minimum and maximum values of PCs 1 and 2. For 3D objects this is one of the most intuitive and visually accessible ways to view the measured shape variation.

To compare the performance of the 3D landmark data to the two commonly used egg shape parameters we calculated elongation and asymmetry from the landmark coordinates. Egg length was calculated as the interlandmark distance between the pole landmarks, and egg width is the interlandmark distance between the two landmarks on the *y*-axis, and elongation is the ratio of these two (length/width). For asymmetry, we used a trigonometric approach to calculate the distance from the lower vertex of the egg (landmark 1) to the point where the polar axis intersects the equatorial axis (where the hypotenuse was the average of the interlandmark distances between the two landmarks on the *y*-axis and the lower vertex).

To test for statistical differences in egg shape between species and between clutches of each species, we used the Procrustes ANOVA evaluated for significance with the F-test (*Goodall, 1991*). This distance-based ANOVA (D-ANOVA) uses Procrustes distances among specimens rather than explained covariance matrices among variables, but is statistically equivalent to a regular analysis of variance and is beneficial for high-dimensional datasets since only the number of individuals is important in the model. We evaluated a nested model of egg shape ~species/clutches. Significance testing was achieved through permutation using a residual randomisation permutation procedure involving 1,000 permutations (*Collyer, Sekora & Adams, 2015*).

To quantify the amount of egg shape variation within-species, as well as within-clutches of each species, we measured the dispersion of all observations around the mean shape for the group. For shape data, this is the Procrustes Variance, which is the mean squared Procrustes distance of each specimen from the average shape, and can be calculated as the sum of the diagonal elements of the covariance matrix of that group (*Zelditch, Swiderski & Sheets, 2012*). To test for statistical differences in disparity between clutches, we calculated absolute differences in Procrustes variances between clutches and used these as test statistics in a permutation procedure, where the Procrustes variances residuals are randomised among groups. For each species, 1,000 permutations were performed.

All analyses were performed in R v.3.2.2 (*R Core Team, 2015*) using the *geomorph* library v.3.0 (*Adams & Otárola-Castillo, 2013*; *Adams, Collyer & Sherratt, 2016*). Digitising routines are provided in the Figshare repository (https://doi.org/10.6084/m9.figshare.3382501.v1).

## RESULTS

Our data indicate a broad range of egg shapes for all species studied (Fig. 3A). The first two PCs accounted for 83.8% of the total egg shape variation across species, and therefore a biplot of the two axes provides a reasonable approximation of the egg morphospace for this study. The remaining PCs each contributed less than 5% of the total variation and are not discussed further. PC1 (73.3%) is associated with elongation of the egg; negative PC1 scores correspond with shorter, squatter eggs, while positive PC1 scores correspond with taller, more slender eggs. Eggs from all species were spread along the PC1 axis, revealing high variability in egg elongation within each species. PC2 (10.5%)

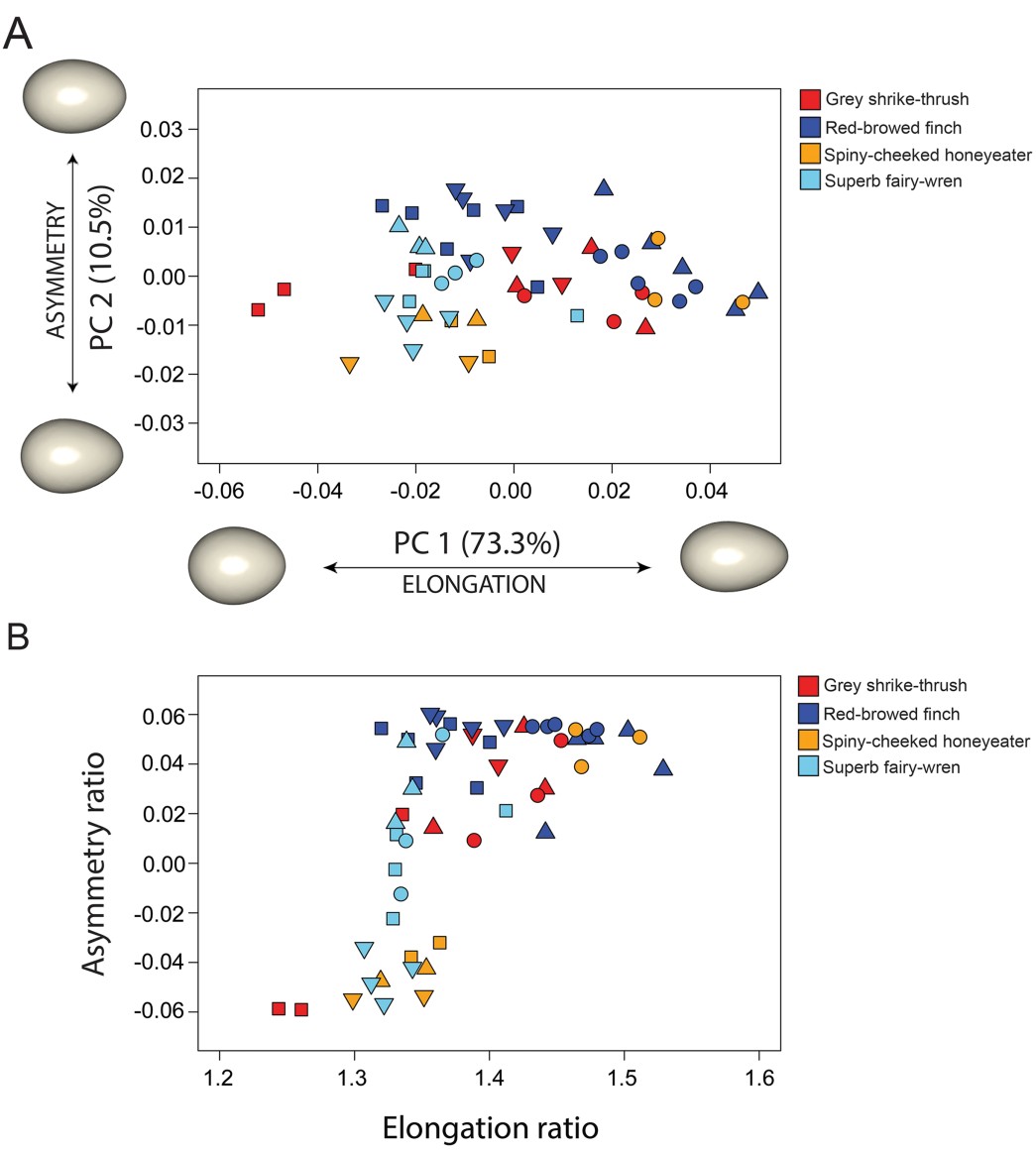

**Figure 3 Comparison between 3D and 2D approach to quantify egg shape variation among four bird species.** Each symbol in shape space represents a single egg. Proximity of each symbol indicates similarity in shape. Symbol colour represents species and shape represents clutch (Table 1). (A) Morphospace defined by the two first principal components (PC's) of shape variance using 3D landmark data. The percentage of total variance described by each axis is shown in parentheses. Shapes associated with the extreme ends of each PC axis are shown as warped surface models (see text for details). The origin point corresponds to the mean shape. (B) Morphospace of egg shape in two dimensions (elongation ratio and asymmetry ratio) using 2D linear measurements.

is associated with tapering in egg shape, so called egg asymmetry, with negative PC2 scores associated with eggs that were more asymmetrical and positive PC2 scores associated with eggs that were more ovoid, with maximum egg breadth only increasing slightly from low to high values of this axis. The two main axes therefore capture the same general features that are commonly studied with 2D approaches. The correlation

**Table 2 Nested D-ANOVA evaluating variation in shape between species and between clutches within each species.**

|  | Df | SS | MS | $R^2$ | F | Z | Pr(>F) |
|---|---|---|---|---|---|---|---|
| Species | 3 | 0.008318 | 0.00277278 | 0.19978 | 1.4768 | 0.4613 | 0.324675 |
| Species: clutch | 12 | 0.022531 | 0.00187758 | 0.54112 | 6.7875 | 6.1576 | 0.000999 |
| Residuals | 39 | 0.010788 | 0.00027662 | 0.25910 |  |  |  |
| Total | 54 | 0.041638 |  |  |  |  |  |

**Note:**
$P$-values based on 1,000 random residual permutations.

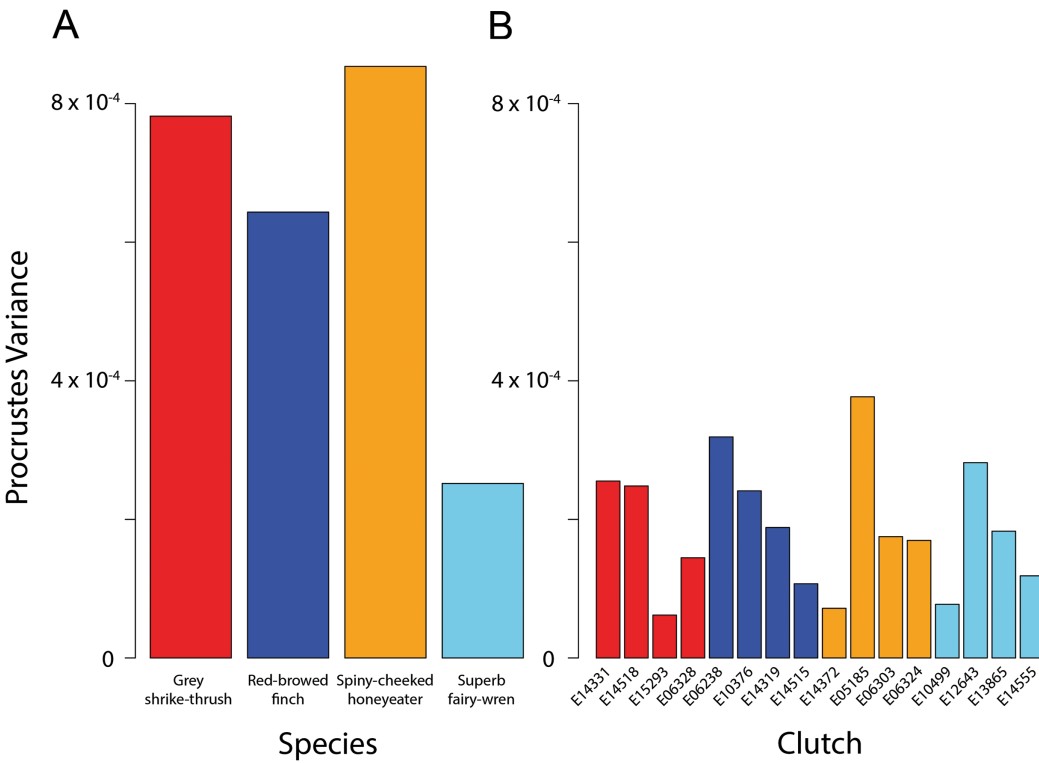

**Figure 4 Bird egg shape disparity within species and clutches.** Disparity, measured as Procrustes variance, of egg shape within (A) species and (B) clutches. Colours correspond to species as in Figs. 1 and 3. The $y$-axes of both graphs are plotted to the same scale. Pairwise comparisons of clutch disparity are presented in Table 4.

between PC1 scores and the parameter of elongation is strong ($r = 0.99$), and the correlation between PC2 scores and the parameter of asymmetry is moderate ($r = 0.66$).

The four species had significantly different egg shapes based on 1,000 iterations (Procrustes ANOVA, $F_{3,51} = 4.248$, $P < 0.001$). For a nested design, there was no significant difference in shape between species when clutches were considered, however clutches within-species are significantly different (Table 2). Figure 3A shows clearly that there is substantial within-species variation in egg shape for all species: within-species disparity (Procrustes variance) is particularly high for the grey shrike thrush, red-browed finch and spiny-cheeked honeyeater (Fig. 4A). The high disparity in egg shape

**Table 3 Pairwise comparisons of Procrustes variance between clutches.**

**Grey shrike-thrush (*Colluricincla harmonica*)**

|  | E14331 | E14518 | E15293 | E06328 |
|---|---|---|---|---|
| E14331 | – | 0.9570 | 0.0859 | 0.3237 |
| E14518 | 7.06E-06 | – | 0.1039 | 0.3766 |
| E15293 | 1.93E-04 | 1.86E-04 | – | 0.5005 |
| E06328 | 1.11E-04 | 1.04E-04 | 8.27E-05 | – |

**Red-browed finch (*Neochmia temporalis*)**

|  | E06238 | E10376 | E14319 | E14515 |
|---|---|---|---|---|
| E06238 | – | 0.4266 | 0.1678 | **0.0250** |
| E10376 | 7.79E-05 | – | 0.5794 | 0.1299 |
| E14319 | 1.31E-04 | 5.28E-05 | – | 0.3766 |
| E14515 | 2.12E-04 | 1.34E-04 | 8.13E-05 | – |

**Spiny-cheeked honeyeater (*Acanthagenys rufogularis*)**

|  | E14372 | E05185 | E06303 | E06324 |
|---|---|---|---|---|
| E14372 | – | **2.50E-03** | 0.3861 | 0.3951 |
| E05185 | 3.05E-04 | – | 0.0919 | **0.0490** |
| E06303 | 1.03E-04 | 2.02E-04 | – | 0.9755 |
| E06324 | 9.80E-05 | 2.07E-04 | 5.47E-06 | – |

**Superb fairy-wren (*Malurus cyaneus*)**

|  | E10499 | E12643 | E13865 | E14555 |
|---|---|---|---|---|
| E10499 | – | 0.0829 | 0.4206 | 0.6523 |
| E12643 | 2.04E-04 | – | 0.4615 | 0.2138 |
| E13865 | 1.06E-04 | 9.90E-05 | – | 0.5495 |
| E14555 | 4.12E-05 | 1.63E-04 | 6.43E-05 | – |

Notes:
Values in the lower triangle are the observed pairwise absolute differences (distances) among clutch Procrustes variances. Upper triangle values are *P*-values associated with pairwise differences (1,000 permutations). *P*-values in bold are significant at the 5% level.

observed within-species is due to between-clutch shape differences as well as within-clutch disparity (Fig. 4B).

Clutch disparity appears quite variable within each species (Fig. 4A), however the Procrustes variances of clutches in all but three pairwise comparisons were not significantly different from each other within each species (Fig. 4B; Table 3). Overall mean clutch disparity was not significantly different among species (ANOVA, $F_{3,12}$ = 0.184, $P$ = 0.906). Together these results indicate that within-species egg shape variation is due to between-clutch differences as well as within-clutch differences in egg shape.

## DISCUSSION

The variety of egg shapes extrapolated in this study concurred with those previously described for each species in the literature. The only shape described for superb fairy-wren eggs is an elongated oval, with markings commonly concentrated at the broader end

(*North, 1889*; *Schodde, 1982*; *Campbell, 1990*), implying that their eggs are only slightly rounder at one end. In contrast, anecdotal accounts refer to several different shapes to describe eggs from the other species included in this study (*Marchant & Higgins, 1990*). The eggs of all grey shrike-thrush subspecies have been described as either oval, rounded oval, stout oval to thick oval, or elongate-oval, with some slightly pointed at each end (*Marchant & Higgins, 1990*). Visualisation of the egg shapes through PCA show that all grey shrike-thrush eggs incorporated in the present study were at least slightly pointed at one pole, with some clutches clearly more elongated than others. Eggs of the red-browed finch encompass a wide range of eggshell shapes (Fig. 3), with this species producing the largest clutch size of the species included in this study (Table 1). Egg shapes of the Pied flycatcher *Ficedula hypoleuca* show high variability within populations, ranging from pointed to ellipsoidal to nearly spherical, and is proposed to be linked to clutch size (*Kern & Cowie, 1996*).

Spiny-cheeked honeyeaters are reported to have oval or elongate-oval eggs (*North, 1889*; *Campbell, 1990*), with spots and blotches usually concentrated at the broader end (*North, 1889*; *Serventy & Whittell, 1962*; *Campbell, 1990*). Only one clutch of eggs from this species (E06324) was very elongated, with all other eggs being relatively squat. This may be attributed to geographic location; E06324 was from New South Wales, while the other clutches used from this species were from South Australia (Fig. 1B). Similarly, the grey shrike-thrush clutch collected from South Australia (E14518) grouped together at the low end of PC1 values (Fig. 3, red square symbol), signifying less elongated eggs compared to the three clutches collected from New South Wales for this species (Fig. 1B). It is possible that difference in climatic conditions between geographic locations may be driving differences in egg elongation in these species. In a study involving 310 native Australian passerines, avian eggs were found to be less elongated in areas that were hot and dry or contained sparse plant canopies (*Englert Duursma et al., 2018*). It was proposed that rounder eggs may be better adapted to hot and dry climatic conditions by facilitating gas exchange between the embryo and the environment. This is because spherical eggs have a lower surface area to volume ratio than ovoid eggs, and thus, may potentially gain and lose heat more slowly, and lose less water. Whether geographical range may be driving variation in egg shape within a species would benefit from additional work to test variability in total shell porosity within a species for different egg shapes, and the influence of egg shape (considering both elongation and asymmetry) on egg surface area to volume ratio.

Surprisingly, a large range of egg shapes was found within each species, resulting from differences among clutches. Whether variation in egg shape between clutches for these species is due to differences inherent in individual breeding females, or the location, climatic condition and year of collection remains unknown. Further information and quantification of impacts of hatch order and geographic location are required to increase our understanding of the processes involved in producing particular shaped eggs. We believe that the 3D methodological approach demonstrated here has the potential to achieve this, allowing the quantification of fine-scale differences in egg shape that may not be discernible using 2D methods. Our method can be applied to any spherical to conical

shaped object, including eggs from various taxa; from birds, and extant reptiles to non-avian dinosaurs and pterosaurs.

The geometric morphometric approach demonstrated here is complementary to standard 2D approaches (*Deeming & Ruta, 2014*; *Stoddard et al., 2017*). We found that the main PC axes summarising the highly multivariate dataset of landmark coordinates, which is a standard tool in geometric morphometrics, discern the primary morphological traits measured in egg-shape research: elongation and asymmetry. Our results indicate that elongation is captured in a similar way by both methods, however asymmetry is less so (Fig. 3). Given the 3D nature of an egg, the way that the equatorial axis is defined in 2D will have important consequences on the estimate of asymmetry. Our method mathematically aligns the 3D egg models by the principal axes prior to digitisation, finding the maximum length, width and breadth of the egg. As such, the landmarks in 3D capture different aspects of egg asymmetry than the traditional asymmetry parameter does. Our study aligns with others that have shown that 3D landmark methods used on highly-3D structures are preferable over 2D projections (*Álvarez & Perez, 2013*; *Cardini, 2014*; *Buser, Sidlauskas & Summers, 2018*).

Our method may also be of relevance to poultry science, enabling the inference of egg quality based on egg geometry parameters (*Narushin & Romanov, 2002*). The physical structure and chemical composition of an eggshell is commonly used as an indicator of egg quality and stability during storage, with significant and direct effects on prices when eggs are graded (*Narushin & Romanov, 2002*). As egg quality traits are associated with hatchability and are moderately heritable, it is important that birds producing eggs with desirable traits are retained for future breeding (*Ragozina, 1961*; *Rose, Simpson & Manning, 1996*). Physical factors generally appreciated in eggs are shell appearance and strength, egg size, weight and the size and appearance of the albumen and yolk (*Murray et al., 2013*). Such information can also be used to provide tangible advice to farmers on the appropriate requirements for good quality eggs. Previous studies have used the less accurate traditional linear measurements to evaluate egg geometry parameters to infer egg quality of domesticated chickens. We believe that volumetric methods such as micro-CT may provide a viable option to provide precise measurements of egg volume and size, and eggshell thickness and shape. Egg quality traits are greatly influenced by breeding and environmental factors (*Clark, Ewert & Nelson, 2001*; *Johnson, Leyhe & Werner, 2001*). As eggs are a primary source of animal protein globally, with levels of egg consumption rising particularly in developing countries (*Kern & Cowie, 1996*), it is important that new scientific approaches are adopted to evaluate egg quality and production.

In biology, palaeontology and medicine, the demand for accurate 3D models of specimens is continuously growing, as is the need to quantify and compare their geometry. Analysis of accurate 3D models using a combination of techniques (e.g. geometric morphometrics and finite element analysis) has enabled researchers to explore the function and evolution of anatomical structures based on morphological differences in their shape (*Polly et al., 2016*). Both volume and surface data are suitable for 3D geometric morphometrics and finite element analysis. Volume and surface scanning

are both non-contact methods, and therefore respond to increasing requirements in conservation, cultural heritage or repatriation programs. The absence of ionizing radiation makes surface scanning a non-destructive/non-invasive measurement tool. Volume scanning (CT, synchrotron, MRI, Terahertz, infrared) captures both the external and internal structure of an object, whereas surface scanning is limited to the outer shell of an object. Surface scanning refers to optical systems that measure objects through visible light (i.e. non-ionising radiation). Photogrammetry and laser scanning are the two main sources that can provide surface data. Low-cost photogrammetric methods are now available for high resolution topographic reconstruction, ideally suited for low-budget research (*Westoby et al., 2012*). Any camera can be used for photogrammetric purposes, depending on the accuracy requirements of the study (*Boehler & Marbs, 2004*). Using a set of images containing a high degree of overlap from a wide array of positions, we have been able to capture the full three-dimensional structure of an egg (see Fig. S1). The camera positions and orientation of these images can be solved automatically using image processing packages (e.g. Agisoft Photoscan, Memento, CapturingReality, ImageModeler, Recap and 123D Catch) and a network of targets which known 3D positions can be assigned to scale the object to its real size (*Westoby et al., 2012*). A big advantage of such packages is their simplified and user friendly interface, aimed mainly for non-photogrammetrists. Photogrammetry provides an economical and efficient alternative to more expensive volumetric imaging techniques such as CT or MRI, especially for capturing simple objects at close range such as eggs. Laser scanners are usually preferred when digitising surfaces of very complex, irregular objects. Laser scanner manufacturers normally provide their own software, though an additional software package may also need to be purchased (*Boehler & Marbs, 2004*). The range of affordable methods now available to capture and analyse 3D data will help facilitate future research on the form and function of biological structures.

## CONCLUSIONS

This study provides the first 3D analysis of egg shape and demonstrates its performance in capturing shape variation at multiple biological levels. Our method uses a semi-automated approach to digitise the entire surface of the eggshell to measure variation in shape between specimens. This approach can be applied to any shaped egg as well as any spherical-to-conical shaped object. The code for implementing this egg analysis technique is written using the R language and the script can be downloaded freely (see Data Availability). Utilising a 3D-GM approach offers an advantage over 2D approaches by improving the accuracy of shape, volume and size measurements. Further research is required to evaluate the degree of error for different 3D and 2D approaches to measure egg shape.

## ACKNOWLEDGEMENTS

We thank Leo Joseph, Alex Drew and Margaret Cawsey from the Australian National Wildlife Collection, Canberra for providing access to specimens and use of the micro-CT scanner. We thank Dr Heinrich Mallison at the Museum für Naturkunde, Berlin, for

sharing his expertise in photogrammetry and Professor Tim Birkhead for providing access to specimens at the Alfred Denny Museum, Sheffield.

### Funding

Marie R.G. Attard was supported by the School of Environmental and Rural Sciences (RE219921 and E239631) at the University of New England and ARC Discovery awarded to Stephen Wroe (DP140102656). The funders had no role in study design, data collection and analysis, decision to publish, or preparation of the manuscript.

### Grant Disclosures

The following grant information was disclosed by the authors:
School of Environmental and Rural Sciences: RE219921 and E239631.
ARC Discovery: DP140102656.

### Competing Interests

The authors declare that they have not competing interests.

### Author Contributions

- Marie R.G. Attard conceived and designed the experiments, performed the experiments, analyzed the data, prepared figures and/or tables, authored or reviewed drafts of the paper, approved the final draft.
- Emma Sherratt conceived and designed the experiments, performed the experiments, analyzed the data, prepared figures and/or tables, authored or reviewed drafts of the paper, approved the final draft.
- Paul McDonald conceived and designed the experiments, authored or reviewed drafts of the paper, approved the final draft.
- Iain Young conceived and designed the experiments, contributed reagents/materials/ analysis tools, authored or reviewed drafts of the paper, approved the final draft.
- Marta Vidal-García performed the experiments, prepared figures and/or tables, authored or reviewed drafts of the paper, approved the final draft.
- Stephen Wroe conceived and designed the experiments, contributed reagents/materials/ analysis tools, authored or reviewed drafts of the paper, approved the final draft.

### Animal Ethics

The following information was supplied relating to ethical approvals (i.e., approving body and any reference numbers):

This study utilises museum preserved bird eggshells held at the Australian National Wildlife Collection, Canberra. The collection manager, Leo Joseph, provided permission to use these eggshells for this study, and no approval number or permits were required. Professor Tim Birkhead provided permission to use eggs from the Alfred Denny Museum, Sheffield for photogrammetry (Fig. S1).

## Data Availability

Micro-CT scans and surface meshes of all specimens included in this study (in DICOM and PLY format, respectively) are available at Figshare: Attard, Marie; Sherratt, Emma (2018): Micro-CT and surface mesh of bird eggshells. figshare. Fileset. https://doi.org/10.6084/m9.figshare.3382477.v1.

The R script developed in this study to characterise egg shape using three-dimensional semilandmark-based geometric morphometrics are available at Figshare: Attard, Marie; Sherratt, Emma (2018): R script for 3D geometric morphometrics of egg shape. figshare. Fileset. https://doi.org/10.6084/m9.figshare.3382501.v1.

## Supplemental Information

Supplemental information for this article can be found online at http://dx.doi.org/10.7717/peerj.5052#supplemental-information.

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
