# Peer review of "A new, three-dimensional geometric morphometric approach to assess egg shape"

_PeerJ, doi:10.7717/peerj.5052_

## Round 0.1 · original submission · Major Revisions

The two reviewers agree that the paper has merit. However, Reviewer 1 wanted more agreement between the title and the stated goal of the paper, among other things (such as additional citations), and reviewer 2 felt that some of the conclusions may be overstated. In particular, sample size may be an issue and you need to either demonstrate that your current sample is unbiased, or refrain from making strong conclusions about morphospace with your given sample size.

·

Basic reporting

• Clear, unambiguous, professional English language used throughout.

OK, some minor suggestions made throughout the text.

• Intro & background to show context. Literature well referenced & relevant.

I really liked introduction section. All background showing the broad importance of this study (from comparative analyses to poultry industry) is in this section.

Experimental design

• Original primary research within Scope of the journal.

Yes, the study is within PeerJ Scope. It presents a new method for accessing egg shape variation. Knowing how egg shape varies from within clutches to across species is very important for comparative analyses (it is also relevant for poultry industry) but has not received enough attention due to difficulties in evaluating it. The method presented in this paper for accessing egg shape variation is potentially important for future studies, but need improvements (see general comments, specially those of lines 129-130 and 392-393).

• Methods described with sufficient detail & information to replicate.

OK. More than that, scripts for analysis routines are also provided.

• Conclusion well stated, linked to original research question & limited to supporting results.

The manuscript lacks a general conclusion. The last paragraph of discussion only concludes about the relevance of this method for poultry industry.
Authors need to include a paragraph summarizing the importance of their method for future studies, based on the last paragraph of the introduction.

Validity of the findings

• Data is robust, statistically sound, & controlled.
It is, but see General Comment of Introduction (lines 129-130).

Additional comments

The manuscript is well written, designed, and organized, presents a new and important methodology for accessing egg shape variation, but need some improvements. The most important are listed below. Small ones are in the annotated pdf (please ignore annotations in blue).

Abstract:
The authors state (abstract, line 40) that: “The approach developed in this study to assess variation in shape is freely accessible…”
Although the method is freely available, this statement may be misleading since the costs for acquiring a 3D scanner can be prohibitively high. Authors should substitute “approach” for “routine” or something similar or compare the costs of their method with other methods for obtaining egg parameters.

Introduction:
In the last paragraph (lines 129-130) authors state: “The aim of this study is to investigate within- and between-clutch variation of avian egg shell shape”, whereas the manuscript title is: “A new, three-dimensional geometric morphometric approach to assess egg shape variation within and between clutches”.
These statements must agree. For me it seems that the study is more about a new method, as stated in the title, but using a “case-study” with a few selected species to present the use of their method. If the focus is on investigating variation in egg shape within- and between-clutch, it seems that data is not robust enough and analyses would require including phylogenetic information. If the focus is on a new method, comparing (e.g. cost, error, etc) with previous 2D methods is lacking in the manuscript and is absolutely necessary.


Methods:
Unable to know the costs and how easy to implement this method.

Lines 207-208: There are other methods, using mathematical equations that should be mentioned here and even implemented in your study:

Preston, F. W. 1969. Shapes of birds’ eggs: extant North American families. - The Auk: 246–264.
Troscianko, J. 2014. A simple tool for calculating egg shape, volume and surface area from digital images. - Ibis 156: 874–878.

Lines 249-256: Authors criticize using 2D egg contours due to possible imperfections on radial symmetry of eggs. I think they should show how much error is associated with “subtle differences in curvature around the entire eggshell” to show the reader how much improvement their 3D method brings compared to 2D methods. Is really misleading assuming radial symmetry?

Lines 281-283: See comment at lines 391-392.

Results:

Lines 391-392: Troscianko (2014) must be cited here. I am sure that it is possible to quantify the same kind of differences in egg shape using his approach, especially because the main sources of variation (PC1 and PC2) were related to elongation and tapering, respectively. Possibly, by using Micro-CT you can find even finer-scale differences, but according to the results they were unimportant in describing egg shape.
I also do not like using Principal Components to describe egg shapes because of repeatability of results (interpretation of PC) in future studies. PCs are used to reduce dimensionality of multivariate data, but their interpretation is usually subjective and depends on the correlations among variables and variable loadings on each PC (Cadima and Jolliffe 1995). Because egg shape may be easily decomposed in specific parameters, I do not see any reason for using PCs. Using specific parameters (see Troscianko 2014, Preston 1969) will make egg shape results comparable across studies.
Given that PC1 and PC2 were related to elongation and tapering, I strongly suggest using: 1) the same (or equivalent) parameters/indexes used in Troscianko (2014), Preston (1969), Havlíček et al. (2008), and/or Narushin (2001); 2) using directly elongation, tapering, radial symmetry imperfections, etc. instead of Principal Components.
It is a shame that Troscianko’s did not implement reporting the value of egg shape parameters, because it does not allow comparing results among studies. Would be very helpful for future studies if you could do that using your approach.

Cadima, J. and Jolliffe, I. T. 1995. Loading and correlations in the interpretation of principle compenents. - Journal of Applied Statistics 22: 203–214.
Preston, F. W. 1969. Shapes of birds’ eggs: extant North American families. - The Auk: 246–264.
Troscianko, J. 2014. A simple tool for calculating egg shape, volume and surface area from digital images. - Ibis 156: 874–878.



Table 1:
Most of table 1 columns (i.e. Order, Clade, Body length, Body mass, Primary diet, Clutch size average, Clutch size range, Nest material, Nest shape, Nest site, Incubation, Incubation period, Young development) are unnecessary since all this information was not was not evaluated in the analyses.

Order: not needed since all species are Passeriformes

Table 3:
Headings are incomplete. It is not possible to know if clutch refers to a unique clutch/set id or to clutch size of that set. Also it is not possible to know what “variation in shape” refers to. Is it a PC1, PC2, or another parameter?

Reviewer 2 ·

Basic reporting

No Comments.

Experimental design

No Comments.

Validity of the findings

In this paper, the authors present a new way of quantifying egg shape based on 3D geometric morphometrics derived from micro-CT data. This method offers some advantages over traditional methods of shape description because it incorporates details about curvature around the entire eggshell. Due to the novelty of this approach, I think this paper works well as a methods paper and for demonstrating proof-of-concept.

Next the authors ask the following:

“With these data, we investigate whether there is greater shape variation within a clutch than between clutches for a given species, and ask if egg shape be used to distinguish between bird species? According to general assumptions in evolutionary comparative approaches across species or taxa, intraspecific variation in egg shape should be lower than interspecific variation (e.g., Felsenstein, 1985). If natural selection is acting on egg shape, we would expect differences in egg shell shape will reflect adaptations that will optimise incubation and/or hatchability (Hoyt, 1976).”

My main concern here is that the authors have a low sample size: 4 clutches only for each of their 4 target species, for a total of 16 clutches. To me, it seems critical to include an analysis that shows the point at which adding additional clutches per species no longer increases the occupied morphospace for that species. In other words, in Figure 3, what happens if you add 1 or 2 (or 3, 4, 5…) more clutches of Grey shrike-thrush? Does morphospace increase or stay the same in size? What is the effect of your sample size on your analyses, and how might low sample sizes lead to bias? Without addressing this, it’s difficult to know whether the observed differences within and between species are unbiased. Do the authors have access to more than 4 clutches per species? Finally, how do the authors intend to test the prediction that “if natural selection is acting on egg shape, we would expect differences in egg shell shape will reflect adaptations that will optimize incubation and/or hatchability”?

The authors write in the discussion that:

“Surprisingly, we find no evidence of selective pressure in the form of a constraint on egg shape for the species investigated. Overlap in shape independent of size across species suggests that either there is little constraint or selection on shape across these diverse taxa, or that shared selective influences have led to convergence.”

How have you demonstrated this? What specific hypotheses were you testing, and how did your tests lead to this conclusion? There might be some "hidden" constraints. All four species are similar in development, incubation and even average clutch size. Perhaps altricial species are, in fact, constrained to your observed egg shape morphospace. What if egg shape is related to clutch size, and selection for eggs of a particular shape is, in fact, strong for clutches of these sizes? With so few species (4) and no phylogenetically controlled statistical models, these statements are a bit of stretch -- and require more explanation.

Additional comments

I think this study would be improved if the authors could address the two points raised above: issues with sample size and treatment of ecological/life history traits in assessing adaptive hypotheses for egg shape. The solution might lie in focusing on the benefits of the shape analysis approach (which is already done to great effect, and could be improved upon by direct comparison to other shape quantification methods) rather than on the study’s ability to resolve questions about adaptive explanations for egg shape variation.

---

## Round 0.2 · Major Revisions

A reviewer has given a thorough reaction to the manuscript. Overall the reviewer finds merit, but notes that 1) the superiority of the 3D method is oversold and 2) there are confusing aspects to Fig. 4 and Table 4 that must be rectified. Overall the promise of the 3D method needs to be toned down throughout the manuscript, especially since comparisons with other methods were beyond the scope of the study. The statements about constraint on egg shape need to be refined as well. I look forward to reading how you substantively addressed all of these points.

Reviewer 2 ·

Basic reporting

See below

Experimental design

See below

Validity of the findings

See below

Additional comments

Understanding egg shape variation is an important and timely topic, and in this revision the authors argue that 3D geometric morphometrics is the way forward, providing “a more precise representation of eggshell curvature.” However, the data provided in this study do not show that the method provides a more precise representation. As the authors write in their response “We recognize the need to evaluate the degree of error for different 3D and 2D approaches to measure egg shape...but unfortunately [that] is beyond the scope of this study.” Therefore, I think the authors can present their method as a promising new tool which could be more precise – but statements like these in the abstract are, in my mind, not supported by the current work.

In fact, as Reviewer 1 noted, the variation detected by PCA still reveals the classic axes of egg shape variation: elongation and asymmetry. It is therefore difficult to know what the advantages of micro-CT/ 3D morphometrics may be over cheaper methods (photography). Because the authors no longer have access to the physical egg clutches used in this study, they cannot provide a comparison of different methods. An additional challenge, raised also by Reviewer 1, is that principal components can be difficult to interpret. In particular, using PCs to characterize a morphospace (see Figure 3) can be problematic, because PCs change depending on the data used. Choosing 4 different species would likely reveal different PCs, so the space is not robust to additional datasets or studies. Why not use specific parameters to define your space, as suggested by Reviewer 1? The authors defend PCA but it seems that nothing would be lost by using the rich morphometrics data to also calculate specific parameters (like elongation and asymmetry).

“As such, we find no evidence of selective pressure in the form of a constraint on egg shape for the species investigated.” I am still not sure what you mean here. The fact that there is egg variation between and within clutches does not show that there are no constraints on egg shape. There likely are real biophysical constraints on egg shape (eggs are not cubes, for example).

Looking at Figure 4, I am surprised to see that the turquoise bars in B (presumably representing Superb fairy-wren) show considerable among-clutch variation, but this does not correspond to the low species-level variance shown for SFW in A. Then I looked at Table 4 and I must confess I am very confused. There seems to be no correspondence between the color of the clutches in B and the clutches shown in Table 4. For example, looking at SFW clutches (which I thought were in turquoise in B), the clutches listed here in Table 4 are E10499 (red in Fig. 4 B), E12643 (red in Fig. 4B), E13865 (blue in Fig. 4B), and E14555 (orange in Fig. 4B). What is going on here? I am not sure how to interpret these data any more.

In sum, I think this paper still has potential, but the focus should be on the possible merits of the new method. I am excited about the new method and I think the case study works well here for demonstrating that it can work well for showing subtle intra-clutch variation. Statements about how this method is more accurate should be tempered, however, given the current data. I think discussing the need for investigating inter- vs. intra-clutch variation is an important one, but I am no longer sure what is being shown in Figure 4B and how that relates to Table 4. Finally, I would remove any discussion of selection pressure on egg shape. With only 16 clutches representing 4 species, your data seem too limited to comment on this.

---

## Round 0.3 · accepted · Accept

Thank you for your revisions on the manuscript.

#